# Death toll among the Bangladeshi refugees of the 1971 war

**Kaustubh Adhikari** [1,2]*, **Nazmul Islam**[3], **Mohammad Mahbubur Rahman Jalal**[4]

**1** Department of Mathematics and Statistics, The Open University, Milton Keynes, United Kingdom,
**2** Department of Genetics, Evolution and Environment, University College London, London, United
Kingdom, **3** Qualcomm Technologies, Inc., Boxborough, Massachusetts, United States of America,
**4** Center for Bangladesh Genocide Research, McKinney, Texas, United States of America

* kaustubh.adhikari@open.ac.uk

## Abstract

Bangladesh achieved its independence in 1971 through a violent liberation war. To avoid persecution by the Pakistani army, 9.9 million Bangladeshi refugees escaped to India. Medicine and food supplies to these camps were not adequate to meet the necessities of such a large population of refugees. Therefore, poor condition of these camps resulted in a higher death rate among the refugees than the peacetime death rate of Bangladeshis. This paper reviews reported death tolls in several refugee camps in India as published in newspapers and peer-reviewed journals. Extrapolating these figures, we estimate the total death toll among the refugees in 1971. We also estimate the overall 'excess death toll', the difference between the actual death toll and the expected natural death toll among these refugees, to be approximately 562,915 deaths. The confidence interval for the estimated excess death toll among the refugees is (323,562, 802,268) deaths.

**Data availability statement:** All raw data required to replicate the results of this study

## Introduction

The West Pakistani army started a violent crackdown, named "Operation Searchlight," on the night of March 25, 1971, to suppress pro-independence movements in East Pakistan [1,2]. The attack marked the beginning of the liberation war of Bangladesh. During this war, almost 9.9 million Bangladeshi refugees escaped to India [3] to avoid being persecuted by the Pakistani army [4], and took shelter in refugee camps across different border states of India. These refugees entered India gradually from March 1971 till December 1971. The war ended on December 16, 1971, through the surrender of the Pakistani army at the hands of Bangladesh-India allied forces. The refugees started to return to Bangladesh in large numbers soon after, and most of the refugees were repatriated by February 1972 [3].

The refugee numbers were so high that they sometimes surpassed the local resident numbers in the areas where they settled [5]. This created a huge strain on the resources of the Indian Government and humanitarian organizations [2]. Inadequate aid and difficult living conditions meant death rates were several times higher than in normal times [6]. Even though Indian records of refugee influx were considered relatively reliable [2], no record could be kept of refugee mortality in these strained circumstances; a report by the US Senate Committee noted that "Accurate statistics are difficult to obtain or even estimate" [7].

are reported in tables 1 & 2 of the main text and in tables provided in S1, S5, S6, S8 Text.

**Funding:** The authors received no specific funding for this work.

**Competing interests:** The authors have declared that no competing interests exist.

In any refugee crisis, mortality rates are usually several times higher than peacetime death rates in the host country or country of origin [8,9]. The common reasons are usually malnutrition, and problems from poor living environment such as overcrowding and lack of sanitation, causing diarrhea, dysentery and epidemics such as cholera, malaria, tuberculosis etc. [8,10,11]. The importance of any factor varies over time and situation, such as varying seasons, especially the rainy season causing high mortality [11].

The paucity of recorded data on refugee mortality makes accurate estimation of the refugee death toll very difficult [9,10]. This study therefore uses a mix of published surveys, official statements, and newspaper reports to tabulate refugee numbers and death counts in camps around India, and projects the death rates to the entire refugee population. When data from multiple sources of various quality are available, it is common to estimate an aggregate by combining the data statistically, along with 95% confidence intervals that give an idea of the range of values possible [12–14].

It is instructive to estimate the 'excess death toll' for a crisis, which is the excess mortality due to the crisis compared to peacetime levels of mortality [12]. Here 'excess death toll' is the difference between the estimated total death toll and the expected natural death toll that would have occurred among the same people during the same period if the 1971 war had not happened.

In this study, we use statistical methods to estimate the natural (peacetime) and actual death toll among the refugee population to deduce the excess death toll along with its confidence interval. We provide a conservatively calculated estimate and explain that our estimate is probably an undercount of the actual refugee death toll.

## Previous estimates of the death toll

Previously there had been several estimates about the total death toll among the 70 million Bangladeshi population during the 1971 war. These estimates range from low figures such as 50,000–100,000 [15] and 300,000 [16] to a high figure of 3 million [17], which is the official estimate provided by Bangladesh. However, analyses of the death toll in the 1971 liberation war are usually confined to killings or violent deaths during the period. Other non-violent mortalities such as epidemics or malnutrition in the war-torn population are often ignored [18], even though mortality rates among internally displaced people are also usually several times higher than peacetime rates [9]. Even more neglected are the mortality records for the 9.9 million refugees who moved to India in 1971. We focus on the death toll among these 9.9 million refugees in this study, who moved out of the war zone of Bangladesh and therefore are usually completely ignored from discussions around the wartime death toll.

During recent humanitarian crises, humanitarian organizations have kept useful records of the refugee death toll [9]. But data for the 1971 refugee crisis is scant. A review looking at the period 1971–85 could not find enough data for the Bangladesh 1971 crisis [10].

A substantial amount of descriptive literature has focused on the sufferings of the Bangladeshi refugees during 1971, based on personal narratives and eyewitness accounts [2,5,19–21]. Although quantitative reports are comparatively scarce, death rates in the refugee camps were widely discussed in contemporary reports to be horribly high. "Of the approximately 1,000 camps along India's 1,350-mile border with East Pakistan, the best are no more than tolerable and the worst are muddy sinkholes where death has a stronger grip than life", [22]. "They are dying in such numbers we can't even keep count", said a relief worker during a cholera epidemic in one of the refugee camps [6]. The director of a large refugee camp near Sabrum in Tripura mentioned in an interview that his greatest need was a crematorium [20]. Lorries were hired just to transport dead bodies from camps [20]. The refugee population consisted of a large number of vulnerable people such as children [23], who more easily succumbed to

the difficult conditions: "Many of the children are wasted by three or four diseases at once." [22,24,25]. Malik interviewed a Save the Children nurse, Bridget Battey, who worked at the 'better conditioned' Salt Lake camp in late November-December 1971 after the abatement of the monsoon, and yet she observed "A lot of children have died since I came, about 15 per cent" [21]

There have been some crude estimates of death toll among Bangladeshi refugees during the war. For example, A news report on 22nd June 1971 quoted on-field workers about an aggregated estimate of death count among refugees, saying "An accepted estimate here is that in the past two months, since the great exodus from East Pakistan began, perhaps 300,000 persons have died of disease and another 300,000 have died of malnutrition." [26]. Shaheed Chowdhury interviewed a former Red Cross worker who served at camps on the eastern border of West Bengal, and estimated at least 300,000 deaths there; in another interview a former government employee in charge of refugee records for West Bengal indicated receiving reports of at least 300,000 deaths till September [27].

Seaman performed a survey on a sample population of 4,770 refugees in the Salt Lake refugee camp [25] and extrapolated his findings to estimate a total death count of 200,000 among Bangladeshi refugees. However, Salt Lake was one of the 'better conditioned' 'demonstration' camps [22], and the survey did not capture deaths due to epidemics such as cholera etc. Therefore, Seaman's extrapolated estimate of a total death count of 200,000 can only be taken as a loose lower bound. For example, in a news report on 25th December 1971, Seaman himself mentioned that in the Salt Lake camp, 6,000 children aged below eight are estimated to have died out of a total population of 31,000; this death rate applied to the total refugee children population would mean 300,000 deaths among the children aged under eight alone [28].

Recently, Rahman attempted a crude adjustment of the Salt Lake camp's extrapolated death toll by adjusting for some of its limitations, such as the study duration not covering the entire crisis period, arriving at a total death toll of 700,000 [29].

However, as we will see throughout this paper, the death toll among refugees varied significantly between monsoon and non-monsoon periods, and among different camps. A reliable estimate of the total death toll cannot be derived by simply focusing on a single refugee camp. No previous work in the literature has estimated the total death toll among Bangladeshi refugees in India by considering death toll among several refugee camps. We consider death toll among seventeen different refugee camps and the variation of death toll between monsoon and non-monsoon seasons. We use these different factors to estimate the total death toll among Bangladeshi refugees during the 1971 war.

## Materials and methods

### Modelling the death rates: the binomial distribution model

We assume that the probability of death of a refugee (the death rate) in a particular camp is $p$ per day, and the total person-time spent by all refugees in that camp is $n$. Let $q$ denote the probability of survival of a refugee in that camp per day, i.e., $q = 1 - p$.

We will use a Binomial distribution to model the refugee death $r$ in the camp, which is a common distribution used to model count data [14,30]. $r \sim \text{Bin}(n,p)$. Then expected (average) value of $r$ is $E(r) = np$. The variance of $r$ is, $V(r) = np(1-p)$ or $npq$.

Usually, $p$ is very small. For example, the crude death rate of 17 per 1000 people per year corresponds to $17/(1000 \times 365) = 0.000046575$ deaths per person per day.

When $p$ is very small, $q \approx 1$. Hence the variance $V(r)$ can be approximated by $np$, which is also the mean. The standard deviation (SD) is the square root of variance.

Let $N$ denote the total person-time spent by all refugees in India, and let $z$ denote the total refugee death toll. $z$ can be modeled as $Nr/n$. Hence, mean of $z$ is, $E(z) = Np$. Its variance is $V(z) = (N^2/n^2)\ V(r) = pN^2/n$.

## Refugee influx and total refugee count in India

Refugee influx numbers were aggregated by the Government of India, and published in [31] (Volume 1, pages 446,461–462 and Volume 2, pages 81–82), also reported in [20] (pages 94–97). The final refugee population totals, on 15th December 1971, as reported by the Government of India, are presented in Table 1.

In addition to the official publications, refugee numbers were reported occasionally in newspapers quoting national or state government officials [23,32,33]. Refugee numbers for various dates collected from all these sources are presented in S1 Text. Fig 1 shows these influx numbers over time.

The conditions in India got worse during the monsoon season due to floods and increased risks of associated diseases like cholera. The cholera epidemic, for example, started in early June [20] following the onset of monsoon, along with a host of other epidemics such as diarrhea, gastro-enteritis, and skin diseases. All over the world refugee camps suffer from higher death rates during the rainy season [11]. Therefore, in the estimation procedure we treat the monsoon period separately. In S2 Text we look at meteorological data for India in 1971 to explain that the monsoon season that year can be taken from June to September (inclusive). This period is highlighted in grey in the influx Fig 1.

Unofficial refugee counts were sometimes reported in the newspapers, which were higher than the official numbers [32]. The official numbers are unlikely to be exact and contain some numerical inconsistencies (S1 Text) as well as possible slight undercount due to babies born at the refugee camps (Section 3.4 of S3 Text). In particular, the official starting date of records in the Government tables, 10th April, excludes reports of refugee influxes between 25th March and 10th April (S1 Text), and therefore reduces the total person-day calculated for the refugee population. But in general, international observers considered India's official estimates to be reasonable based on personal inspections [2,34,35] (S4 Text).

## Estimating the expected natural death toll among refugees

Soon after the war ended in mid-December, the Indian government announced their intention to facilitate the repatriation of refugees back to Bangladesh. According to UNHCR

**Table 1. Refugee population in India as of December 15, 1971.**

| State | Number of Camps | Number of refugees in Camps | Refugees With Host Families | Total Number of Refugees |
|---|---|---|---|---|
| West Bengal | 492 | 4,849,786 | 2,386,130 | 7,235,916 |
| Tripura | 276 | 834,098 | 547,551 | 1,381,649 |
| Meghalaya | 17 | 591,520 | 76,466 | 667,986 |
| Assam | 28 | 255,642 | 91,913 | 347,555 |
| Bihar | 8 | 36,732 | – | 36,732 |
| Madhya Pradesh | 3 | 219,298 | – | 219,298 |
| Uttar Pradesh | 1 | 10,169 | – | 10,169 |
| **Total** | **825** | **6,797,245** | **3,102,060** | **9,899,305** |

Aggregating reports by the Government of India [31] (Volume 1, page 446,461–462 and Volume 2, page 81–82).

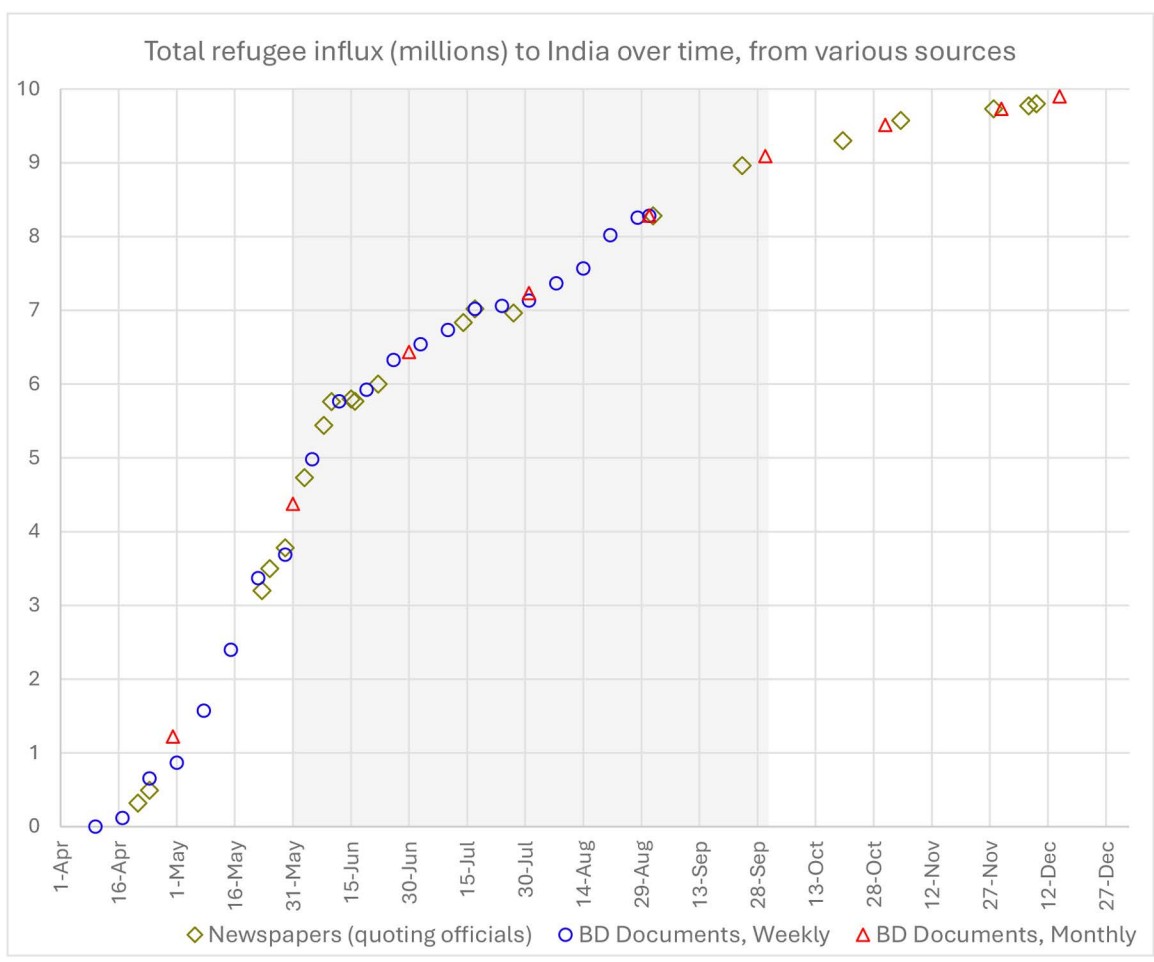

**Fig 1. Total influx of Bangladeshi refugees to India over time in 1971.** The monsoon period of June to September is highlighted in grey. The three different types of data sources (presented in S1 Text) are shown in three different colours and symbols. 'BD Documents' refers to the publication by the Government of India [31].

reports [3], by 6th January 1972 more than a million refugees had already left India, which increased further to 6 million by the end of January and 9 million by the end of February. By next month only 60,000 were remaining in India. A government official was quoted as saying "Immediately after the independence of Bangladesh the refugees just melted away and repatriated of their own will" [5].

Considering that there was very little refugee influx in December since the war ended, and also the death rate going down by that time as winter had set in and the diseases abated, we consider December to be the end-point of our calculations.

The crude death rate (CDR), the number of people dying per 1000 people in a year, is the most standard statistic used to represent death rates [14,36]. It is the most commonly available statistic, as it is an aggregated estimate over all people, so it is possible to calculate even when detailed data across strata, such as by sex or by age groups, is not available.

To calculate the baseline natural death toll, we use the estimated annual crude death rate (CDR) of 17 per 1000 people based on the nationwide estimate for 1971 in Bangladesh [18]. Assuming the same baseline death rate holding true for all Bangladeshi

refugees who moved into India throughout the year 1971, we perform a weighted average based on the duration that they spent as refugees in India. Taking a piecewise sum of the influx numbers (Fig 1), the total person-time spent by all refugees in India is 1853.65 million people × day (S1 Text).

The CDR of 17 per 1000 people per year corresponds to 17 × 1000/365 = 46.575 deaths per million people per day, which can be considered as the binomial probability of death or death rate (following the mathematical notation earlier).

Multiplying this by the total person-time spent by all refugees in India, we estimated the natural death toll among the refugees, following the binomial model.

Under the binomial model (with approximation for small p), this is also the variance of the estimate.

## Age stratification and inflation in variance

The demographic composition of a refugee population is not homogeneous – the population can be divided into several strata such as children, elderly etc. who have substantially disparate mortality risks (S5 Text). Consequently, death rates have large variation across age groups, the infants and children being the most vulnerable and having much higher mortality rates [10,11]. For the 1971 refugees, we can observe the same from a survey on Salt Lake camp [25] (S5 Text).

When age-stratified mortality data is available (S6 Text), this information can be used to obtain a better estimate of the variance (or SD), than one given by the simple binomial formula above. Taking the ratio of the stratified SD to the binomial model SD, we obtain an inflation factor (a multiplying factor), which can be multiplied to any model-based estimate of SD to obtain a better (larger) estimate of SD So, we decide to adjust our SD estimates with this inflation factor to account for this stratification in mortality rates across strata of age.

## Parameters

For the refugee camps, we consider two death rates. One is the baseline death rate among all refugees due to various general causes, which is denoted by $b$.

The second is the increase in death rate during the monsoons, from factors such as epidemics, which is explained in the section on refugee influx. The monsoon season in India was June-September in 1971 (S2 Text). This rate is denoted by $m$. Therefore, during the monsoon period, the total death rate was $b+m$.

Both are measured in the same unit as the CDR discussed earlier, which is the number of deaths per million people per day.

## Aggregation of all refugee camps' death tolls

Table 2 shows the death toll and corresponding information of the refugee camps whose data were aggregated for this study in S7 Text. Fig 2 shows the location of these refugee camps on a map. Instances where deaths were entirely caused by major monsoon-specific death factors such as cholera are marked as '(monsoon)', indicating that only the death rate parameter $m$ (increase in death rate during the monsoons) is relevant here. Deaths attributable to only general causes, i.e., the baseline rate (parameter $b$), are marked as 'General deaths only', even when they occurred during the monsoon period. Instances where both the baseline rate and the monsoon-specific elevated rate (for the entire duration or some part) were applicable are marked as 'All deaths'.

**Table 2. Reported death tolls in different refugee camps that are used in this study.**

| Camp | State | Total population | Total death | Period of consideration for the death count | | | Comment | Source |
|---|---|---|---|---|---|---|---|---|
| | | | | Date Start (1971) | Date End (1971) | Duration (days) | | |
| Salt lake | West Bengal | 170,000 | 3,671 | 01/07 | 30/11 | 153 | General deaths only | [25] |
| Salt lake | West Bengal | 170,000 | 1,250 | 01/06 | 08/06 | 8 | Cholera deaths only (monsoon) | [2, p. 148] |
| Itkhola | West Bengal | 18,000 | 17 | 01/06 | 12/06 | 12 | General deaths only | [23, p. 58] |
| Karimpur | West Bengal | 15,000 | 700 | 01/06 | 17/06 | 17 | Cholera deaths only (monsoon) | [6] |
| Boyra | West Bengal | 40,000 | 5,000 | 01/06 | 01/08 | 62 | Cholera deaths only (monsoon) | [37] |
| Kalyani | West Bengal | 50,000 | 500 | 01/06 | 30/06 | 30 | Cholera deaths only (monsoon) | [20, p. 92] |
| Nadia district | West Bengal | 250,000 | 1,000 | 01/06 | 05/06 | 5 | Cholera deaths only (monsoon) | [33, pp. 119–120] |
| Barasat jail compound | West Bengal | 3,000 | 166 | 25/03 | 27/10 | 217 | All deaths | [38, p. 834] |
| Barasat hospital | West Bengal | 7,000 | 1,250 | 01/07 | 25/07 | 25 | All deaths | [39] |
| Banjetia and Lalbagh | West Bengal | 20,000 | 1,000 | 01/06 | 16/12 | 199 | All deaths | [5, p. 218] |
| Balat & Mailam | Meghalaya | 171,000 | 2,000 | 15/09 | 30/09 | 16 | Cholera deaths only (monsoon) | [2, p. 108] |
| My Long (Myilliem) | Meghalaya | 80,000 | 3,500 | 01/05 | 30/06 | 61 | Cholera and pneumonia (monsoon) | [20, p. 77] |
| Ampati | Meghalaya | 45,000 | 3,500 | 01/05 | 16/12 | 230 | All deaths | [40] |
| Chapor | Assam | 50,000 | 500 | 01/05 | 16/12 | 230 | All deaths | [5, p. 221] |
| Jambu Island | Orissa | 33,000 | 5,000 | 29/10 | 06/11 | 9 | Cyclone deaths only (monsoon) | [41] |
| Mana | Madhya Pradesh | 137,000 | 80 | 21/06 | 03/07 | 13 | General deaths only | [42] |
| Mana | Madhya Pradesh | 60,000 | 819 | 01/09 | 21/09 | 21 | Cholera deaths only (monsoon) | [23, p. 109] |

## Estimating total death toll among refugees

We use the records aggregated in Table 2 to estimate the total death toll among refugees, following the same binomial model which was used to calculate the natural death toll.

There are several limitations to the collected data, for example, no records of mortality rates are available for people living outside the camps, who consisted of one-third of the total refugee population (Table 1). These limitations are discussed later to argue why the current estimate is likely to be an undercount.

## Data points and relevant death rates

The comments in Table 2 explain which death rate is relevant to which data point. For example, the information about the Salt Lake camp in row 1 corresponds to general deaths only, so the applicable death rate will be $b$. On the other hand, the information about the Salt Lake camp in row 2 is only about cholera deaths in the monsoon season, so the applicable death rate will be $m$.

The data for Banjetia and Lalbagh in row 10 spans from 1st June to 16th December, covering 199 days in total, out of which 121 days were within monsoon (1st June to 30th September). Therefore, the increased death rate $m$ was applicable for 121/199 = 0.608 part of the total duration. Hence, the total applicable death rate is $b + 0.608 \times m$.

## Parameter estimation

The unknown parameters in this model are $b$ and $m$, which can be estimated by constructing a set of equations, and solving them via linear regression [30,45].

For example, suppose the information about the Salt Lake camp in row 1 was the only available data. We explain above that the applicable death rate here is $b$. Therefore, this information can be used to build an equation for $b$ and get its estimate.

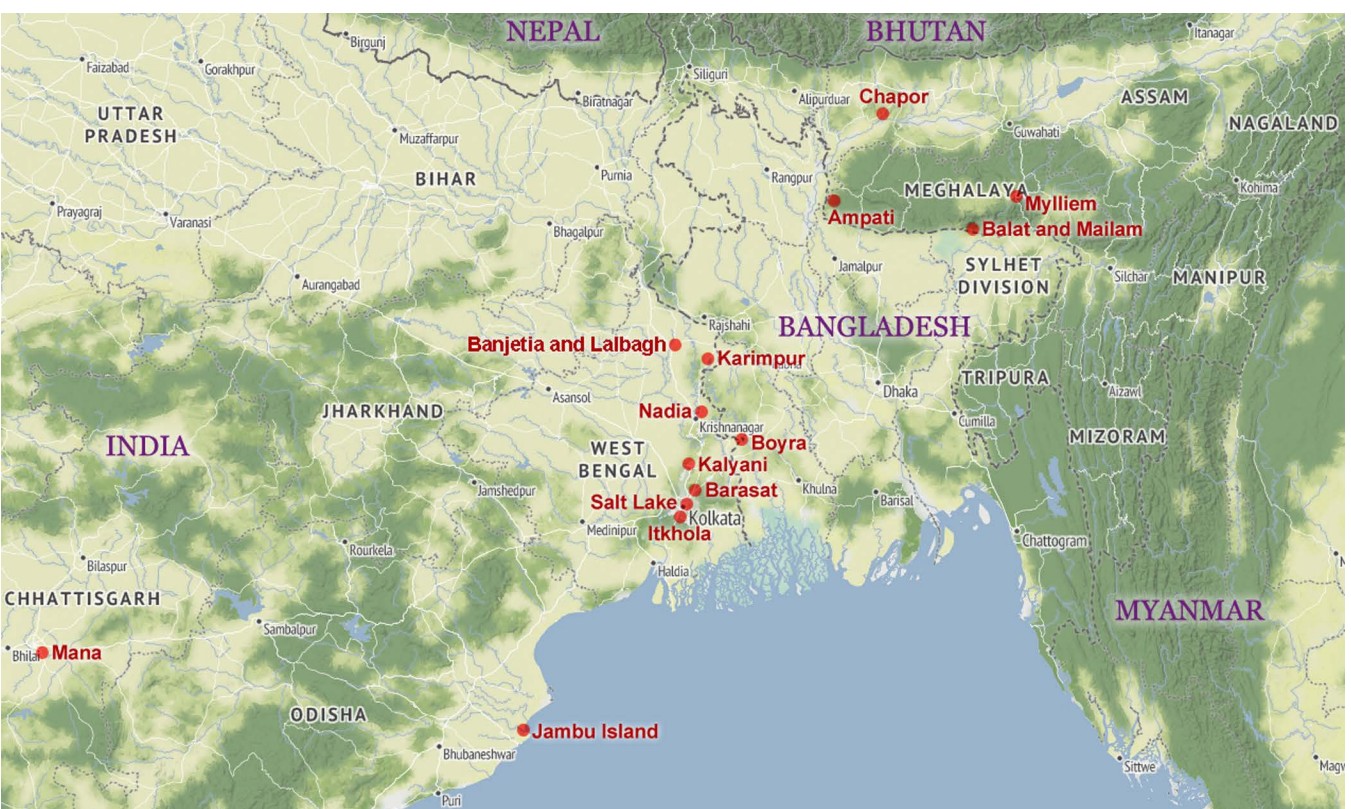

**Fig 2. Location of refugee camps across India that are used in this study.** The locations of the camps are shown as red points, and their names in red text. The names of countries are shown in purple capital letters. Map generated using the package leaflet [43] in statistical software R. Geographic data was drawn from OpenStreetMap [44] within the leaflet package, showing a physical geography (terrain) layer along with the name of major towns, states, and various borders.

The total population of this camp was 170,000, and its duration was 153 days, from 1st July to 30th November. This corresponds to a total person-time of $170,000 \times 153/1000000 = 26.01$ million people × day. It is likely that all the camps started with a smaller population and refugees came in over time, so using the final refugee population for the entire duration overestimates the total person-time, and therefore underestimates the death rate when used in the denominator.

In this camp, 26.01 million people × day were exposed to the death rate $b$ for this period, and the total death was calculated to be 3,671 from (Seaman, 1972). Therefore, the equation for $b$ is $26.01 \times b = 3,671$.

Solving this, the estimated death rate is $b = 3,671/26.01 = 141.14$ deaths per million people per day, or 51.52 deaths per 1000 people per year. This is substantially higher than the CDR for natural deaths discussed earlier, which was 46.575 deaths per million people per day, or 17 deaths per 1000 people per year. Thus, we can see that even for a 'model' camp like Salt Lake, the baseline death rate was substantially higher than the natural death rate.

If data from only a single camp was available, the simple equation as above would suffice to estimate a parameter. However, as data from multiple refugee camps are available, the information needs to be aggregated via a statistical technique. The two parameters were estimated using linear regression, a standard statistical estimation technique [30,45].

The entire set of equations, as constructed in the above example for Salt Lake, for the 17 data points can be written as a linear regression problem, with the two coefficients being $b$ and $m$. This linear regression system can be solved to estimate $b$ and $m$.

As there is considerable variation between the data points, both in terms of their total contribution of person-time, and their death rates, the statistical procedure of robust linear regression [46] was used, as implemented in the function 'rlm' in package 'MASS' for statistical software R. The set of equations (involving the parameters $b$ and $m$) was solved by fitting a robust linear regression model on these data, without intercept, to provide robust estimates of the two parameters.

## Estimate of total death toll

The total death toll can be estimated by aggregating the contributions of the baseline death rate (excluding monsoon-specific factors such as epidemics) and the increased rate due to monsoon. The estimated death rates are multiplied by the corresponding person-times to obtain the estimated number of deaths, from baseline and due to monsoon. These are added to calculate the total death toll.

## Variance of total death toll

Under the binomial model, the above estimate is also the variance of the total. However, as there is considerable variation in the data, we also need to calculate the excess variance due to the variability in the data. If a single parameter was used, this variability would be reflected in the standard error of the parameter, obtained from the regression model. But considering that there are two different parameters which contribute in a specific way to the total estimate, the variability due to the data is better calculated through a leave-one-out bootstrapping procedure [45]. In this procedure, one data point is left out of the dataset, the regression model is fit to estimate the parameters, and the parameters are used to obtain the total death toll. This is done for each of the data points to obtain a set of 17 estimates of total death toll. The variance of these numbers represents the variability due to variations in the source data. The two variances can be added to estimate the total variance.

## Results

### Expected natural death toll among refugees

Multiplying the death rate of 46.575 deaths per million people per day by the total person-time spent by all refugees in India, the estimated natural death toll among the refugees is 1853.65 × 46.575 = 86,334.46 people.

Under the binomial model, the standard deviation (SD) is √86,334.46 = 293.83 deaths.

The inflation factor for standard deviations due to age stratification was calculated to be 1.545 (S6 Text). So, the adjusted standard deviation (SD) is 293.83 × 1.545 = 453.96 deaths.

The width of the 95% confidence interval [30] is given by 2 × SD = 907.92 deaths.

Therefore, the confidence interval for the estimated natural death toll among the refugees is (85,426.54, 87,242.38) deaths.

### Estimated parameter values of the linear model

The estimate of the baseline death rate is:

$b$ = 174.03 deaths per million people per day, or 63.52 deaths per 1000 people per year.

This rate is a bit higher than the rate estimated for the Salt Lake camp above (51.52 deaths per 1000 people per year), which is expected as the Salt Lake camp was a 'model' camp and therefore its death rate was generally lower than the other camps.

The increased death rate due to monsoon is estimated to be:

$m$ = 369.51 deaths per million people per day, or 134.87 deaths per 1000 people per year.

So during the monsoon period, the peak death rate (CDR) was 63.52 + 134.87 = 198.39 deaths per 1000 people per year.

The death rate ratio [9] between crisis and peacetime rates in this case is:

When the baseline rate is applicable, 63.52/17 = 3.74.

When the peak rate was applicable, during the monsoon, 134.87/17 = 11.67.

## Estimate of total death toll

Estimated number of deaths from the baseline death rate is 1853.65 × 174.03 = 322,584.50.

Estimated number of additional deaths from the increased rate due to monsoon is 884.06 × 369.51 = 326,664.90.

The total death toll is: 322,584.50 + 326,664.90 = 649,249.40 deaths.

If the estimate of baseline death rate from Salt Lake camp [25] is used, the estimated number of deaths due to baseline is 1853.65 × 141.14 = 261,620.72. This is similar to the extrapolated estimate of a total death count of "about" 200,000, as suggested in that study [25]. Though Seaman presented a different, higher estimate in an interview: In a news report on 25th December 1971, Seaman explained to the reporter that in the Salt Lake camp, 6,000 children aged below eight are estimated to have died out of a total population of 31,000, which extrapolated to the total refugee children population would mean 300,000 deaths among the children aged under eight alone [28].

As Salt Lake was a 'model' camp [22] with comparatively lower death rates, these estimates compare reasonably to the estimated number of deaths in the entire refugee population from general causes, as obtained above from the baseline rate, 322,585 (rounded).

When a roughly equal number of deaths due to monsoon is added to the baseline estimate, the estimated total mortality becomes 649,249 (rounded). This also compares reasonably with the estimated mortality as mentioned in a news report on 22nd June 1971, which quoted that approximately 300,000 people have died from various diseases, and an equal number have died from malnutrition, making the total estimate by 22nd June to reach 600,000 already out of 5 million refugees [26].

## Variance of total death toll

Using a leave-one-out bootstrapping procedure [45], the variance obtained, combined across the two regression parameters, was 5,999,387,340.43. Adding this to the binomial variance (equal to the estimated total), we obtain the total variance to be 649,249.40 + 5,999,387,340.43 = 6,000,036,589.83.

## Estimate of excess death toll

The estimated total death toll among the refugees during their entire stay in India was 649,249.40 deaths.

The estimated natural death toll among the refugees during the same period was 86,334.46 deaths.

Therefore, the excess death toll among the refugees during this period is 649,249.40 – 86,334.46 = 562,915 deaths (rounded).

The annual crude death rate includes deaths due to all reasons during the year, and would therefore include deaths due to monsoon and epidemics as occurring naturally during the year 1971 [18]. Therefore, the estimated excess death is specifically due to higher-than-normal mortality rates in the refugee population.

## Variance of excess death toll

The estimated variance of the total death toll was 6,000,036,589.83.

The estimated variance of the natural death toll, without adjusting for inflation due to strata-specific variance, was 86,334.46.

As the two quantities were estimated independently, the variance of excess death toll, which is the difference of the two quantities, is the sum of the two variances.

Therefore, the estimated variance of the excess death toll, before adjustment, was: 86,334.46 + 6,000,036,589.83 = 6,000,122,924.29.

The estimated standard error of the excess death toll is the square root of this variance: 77,460.46 deaths.

The inflation factor for standard deviations due to age stratification is 1.545 (S6 Text). Therefore, the adjusted standard deviation (SD) is 77,460.46 × 1.545 = 119,676.41 deaths.

The width of the 95% confidence interval on each side of the estimate is given by 2×SD, which in this case is 2 × 119,676.41 = 239,352.82 deaths.

Therefore, the confidence interval for the estimated excess death toll among the refugees is (562,914.94 – 239,352.82, 562,914.94 + 239,352.82) = (323,562, 802,268) deaths (rounded).

## Why our estimate of death toll is likely to be an underestimate

During various steps taken while constructing the estimate, we explained how we made conservative decisions in each of those steps, e.g.,: (I) not considering 25 March – 10 April within the period of refugee influx, following the Official tables of Government of India, (II) using the highest recorded camp population as the denominator for the entire duration of the camp, or (III) constructing the period of consideration in each camp to be the broadest possible when dealing with incomplete information.

During crisis situations, official death numbers from refugee camps are often underreported for various reasons [8], and the same was concluded for the 1971 camps by observers [25] who suggested that many deaths went unrecorded (S8 Text). As several of the data entries in Table 2 are based on official records, it is likely that true counts would've been higher.

We list below some additional reasons why our estimate is likely to be an undercount, which are explained in detail in S3 Text.

- Death rates in the Northeastern states' refugee camps were probably higher than those in West Bengal's refugee camps.

- Salt Lake was one of the 'better conditioned' refugee camps.

- Higher mortality among refugees living outside camps.

- Undercount of deaths from a specific factor, malnutrition.

- Lack of records on births among refugees after arrival.

- Non-recording of wayside deaths of refugees in India.

- Higher refugee mortality in Burma.

S9 Text present some other information about mortality in various refugee camps which could not be used in the analysis.

## Demonstration of a statistical adjustment for underestimation

In S8 Text we provide reports that the official death counts were likely undercounted, and provide data, specifically in the context of cholera mortality in camps, to show that unofficial death counts were higher than official death counts. In this study we took the conservative route of using only the official counts when both were available, as some readers may feel the unofficial counts as less reliable. Nonetheless, we demonstrate here that this specific cause of

undercounting can be remedied when sufficient data is available, by calculating an adjustment factor between the official and unofficial estimates. As derived in S8 Text, the inflation factor (a multiplying factor) to update the official death counts is estimated to be 2.489.

For example, according to official reports by Government of India [2], there were 1250 deaths in the Salt Lake camp during June from cholera (row 2 of Table 2). Using the inflation factor, the adjusted death count will be 1250 × 2.489 = 3111 (rounded). Similarly, the adjusted death counts in Kalyani, Nadia district, Banjetia and Lalbagh, and Chapor camps will be 1,245, 2,489, 2,489, and 1,245 respectively. When these numbers are used in the statistical calculations, the estimated numbers are slightly higher. The estimated parameters $b$ and $m$ become 73.86 and 151.64 deaths per 1000 people per year, respectively. The peak death rate (CDR) becomes 225.5 deaths per 1000 people per year. The death rate ratio [9] between crisis and peacetime rates, in this case, becomes 4.34 during baseline and 13.26 during peak.

The estimated total death toll becomes 742,398.95 deaths, so the excess death toll would be 656,065 deaths (rounded), which is an increase of 93,150 deaths from the unadjusted figure. The confidence interval for the estimated excess death toll becomes (378,674, 933,455) deaths (rounded). While these numbers are provided as a demonstration of correcting only one aspect of the various reasons for underestimation, throughout the paper we consider only the unadjusted, conservative figures.

## Comparison to estimated death rates in other refugee crises around the world

Keeping accurate records for humanitarian crises is difficult [9,10], as resources are under strain due to the emergency. As the refugees are not part of the resident population, the host country generally excludes them from any demographic surveys such as the census, which are usually the only sources of large-scale systematic records [9]. Mass movement of people also means that numbers change regularly, so estimates might not be consistent or stable over time. And there is often deliberate over- or under-reporting of death due to various reasons (S8 Text) [8]. Still, in recent humanitarian crisis situations, better records are being kept, especially by humanitarian organizations, as it helps them to plan aid provision, estimate effectiveness, and identify areas needing attention. For example, the CEDAT database provided CDR estimates in 1759 cases worldwide for events during 1998–2012 [9].

However, data for earlier refugee crises were much scarcer. A review looked at 11 such situations around the world during 1971–85, but could obtain sufficient usable data for only 3 cases, and could not find enough data for the Bangladesh 1971 crisis [10].

Mortality figures in nearly all cases are estimated from a handful of surveys and reports, often based on a single source, and with small sample sizes. The variability of such estimates is therefore quite high, although estimates of variability or confidence intervals are usually not provided [9].

Death rates observed in the other studies are in line with the rates estimated here, namely 63.52 deaths per 1000 people per year at baseline, and 198.39 during peak. For example, annual CDR observed by [10] ranged from 170 to 406 (deaths per 1000 people per year) in various countries during 1971–85, while at peak times the rate was more than 500. Infant mortality rates reached much higher – in a Sudanese camp the death rate among children younger than 5 years rose to 1113 (deaths per 1000 children per year, during a peak month) due to malnutrition [10]. The refugee crisis in Eastern Sudan during 1985–90 also saw peak rates ranging from 365 to 511 [11]. In recent times, the refugee crisis in Darfur saw rates as high as 120 when a conservative statistical estimation procedure was used, while uncorrected death rates from original sources could be higher than 250 [12].

From the CEDAT database, [9] observed that the death rate ratio between emergency and peacetime rates can be as high as 20 or 50-fold. In our estimation, the death rate ratio was 3.74 at baseline rate and 11.67 during peak, again comparing reasonably.

It can be argued that in more recent times the death rates in emergencies have gone down by some extent due to the availability of better medical interventions among other reasons [10], just as the peacetime death rates have gone down across the world too [18].

## Limitations of the study

The primary limitation of this study is the paucity of source data, which is a common problem in studying humanitarian crises [9,10]. While camp officials and workers were struggling to even provide necessities and medical attention to the refugees, hardly any records were kept of mortalities. The data used in this study comes from three main types of sources: surveys conducted on individual camps, news reports from camps during their operation, and reports published after the war containing personal experiences or interviews. In addition to books and reports (by UNHCR etc.), we combed through the historical newspaper archives of international newspapers (via ProQuest etc.) as well as collections of Indian and Bangladeshi newspapers ([23,32,33] etc.) for 1971–72.

Despite being mostly first-hand accounts or official statements, the records used here are imperfect, and are likely to be approximate figures. They often span a short period or focus on one primary cause of mortality. We have discussed in detail the various limitations and the corresponding assumptions that we make, especially while using incomplete records. We tried to be conservative while making any assumption about the data, such as constructing the period of consideration in each camp, and explained why our current estimate is likely an undercount of the true mortality. Still, we compare our estimate with various other estimates to show that they are of a similar order of magnitude. Researchers with better access to national and state-level newspaper archives in India for 1971–72 might be able to find more first-hand data points and improve the work.

Another limitation of the study is the lack of information stratified by age. Mortality rates varied enormously among different age groups, and the composition of the refugee population differed at different times and places. However age-stratified mortality data was not available beyond the two small surveys [25,38], and the age composition of the refugee population was not known either. If such information were available, the estimate could've been improved. Age and gender stratified data are generally key in the field of demography for studying mortality rates, but unfortunately, such sophisticated methods are inapplicable here due to the paucity of data. If better records become available in the future, better estimates could be obtained.

## Conclusion

Analyses of the death toll in 1971 liberation war usually confine themselves to killings or violent deaths during the period. Non-violent mortalities such as epidemics or malnutrition in the war-torn population are often ignored [18]. Internally displaced people also suffer mortality rates several times higher than peacetime rates [9]; for example, many died inside Bangladesh on their way to or while returning from India, both on the road and in refugee transit camps (S10 Text). Even more neglected are the 9.9 million refugees who moved to India in 1971 in search of some temporary respite from being persecuted by Pakistani army.

We show in our article that, like the people living in Bangladesh during 1971, these refugees too suffered elevated mortality, due to epidemics such as cholera that ravaged relief camps during the difficult living conditions, and since medicine and food supplies to these camps

were not adequate to meet basic necessities of such a large population of refugees [2], so the conditions in the camps became worse than in the general population, with especially severe suffering caused by flood and epidemics [22,25]. Rummel too included the few cholera death reports in his quantitative analysis of the total wartime death toll [1], arguing "Malnutrition, disease, and exposure deaths among the refugees constituted democide. These deaths resulted directly from these pitiful people, largely Hindus, fleeing for their lives before the murderous Pakistan Army" (S4 Text). Working with death toll reports from several refugee camps of West Bengal and North-Eastern states, we estimate the excess mortality in Indian refugee camps during the 1971 war to be more than 562,000. Due to the large variability in source data, we have a relatively large confidence interval of the excess death toll, ranging between 323,562 and 802,268 deaths.

Previously there had been several estimates about the total death toll among the 70 million Bangladeshi population during the 1971 war. These estimates range from low figures such as 50,000–100,000 [15] and 300,000 [16] to a high figure of 3 million [17], which is the official estimate provided by Bangladesh. Unfortunately, most of these estimates are 'guesstimates' [47], i.e., not based on proper surveys but simply on personal accounts or guesses from residents or government officials, making them less reliable. The only quantitative analysis was conducted by Rummel, who aggregated all available estimates to arrive at a figure of 1.5 million [1]; however, the guesstimates aggregated and used in that analysis were unreliable to begin with. In contrast, our estimates of death toll among 10 million refugees are based on direct on-field surveys from relief camps, records from government officials, relief workers, and international relief organizations, which are combined through established statistical procedures.

We argue that any estimate of wartime mortality must include these refugee death toll figures, as these refugees are direct consequences of the war itself, even though they were not residing in the direct war zone. Non-violent deaths in humanitarian crises, such as among internally displaced people and refugees, often outnumber violent deaths, and they should also be attributed to the conflict that was at the root of this displacement [9]. Therefore, our conservatively calculated estimate of excess refugee death – 5.6 lakhs (0.56 million; 1 million is equal to 10 lakhs) among roughly 10 million refugees – refutes the low estimates such as 0.5, 1 or 3 lakhs (0.05, 0.1, or 0.3 million) for the total wartime death toll among the 70 million population.

## Supporting information

**S1 Text. Refugee influx over time.**
(DOCX)

**S2 Text. The monsoon season in India during 1971.**
(DOCX)

**S3 Text. Reasoning why our estimate of death toll is an undercount.**
(DOCX)

**S4 Text. The counter-narrative by Pakistan disputing Indian refugee count.**
(DOCX)

**S5 Text. Refugee population composition and mortality rate by age.**
(DOCX)

**S6 Text. Adjustment for variance inflation due to stratification on age.**
(DOCX)

**S7 Text. Descriptions of refugee camps' death tolls.**
(DOCX)

**S8 Text. Potential adjustment of official death counts.**
(DOCX)

**S9 Text. Other incomplete information about refugee deaths that could not be included in the analysis.**
(DOCX)

**S10 Text. Mortality among refugees within Bangladesh while journeying to and from India.**
(DOCX)

## Acknowledgments

We thank Mr. Mofidul Hoque, co-founder and trustee of the Liberation War Museum of Bangladesh, for providing helpful suggestions. We also thank Dr. Nafisa Tanjeem and Ms. Mowmita Basak Mow for helpful feedback. Thanks also to the reviewers whose suggested corrections have improved the manuscript.

## Author contributions

**Conceptualization:** Kaustubh Adhikari, Nazmul Islam, Mohammad Mahbubur Rahman Jalal.

**Data curation:** Kaustubh Adhikari, Mohammad Mahbubur Rahman Jalal.

**Formal analysis:** Kaustubh Adhikari.

**Methodology:** Kaustubh Adhikari, Nazmul Islam.

**Visualization:** Kaustubh Adhikari.

**Writing – original draft:** Kaustubh Adhikari, Nazmul Islam.

**Writing – review & editing:** Kaustubh Adhikari, Nazmul Islam, Mohammad Mahbubur Rahman Jalal.

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
