## [Decision Letter · Decision Letter 0]

28 Jan 2025

PONE-D-24-42984Death Toll among the Bangladeshi Refugees of the 1971 WarPLOS ONE

Dear Dr. Adhikari,

Thank you for submitting your manuscript to PLOS ONE. After careful consideration, we feel that it has merit but does not fully meet PLOS ONE’s publication criteria as it currently stands. Therefore, we invite you to submit a revised version of the manuscript that addresses the points raised during the review process.

Please submit your revised manuscript by Mar 14 2025 11:59PM with the consideration and carfully revised according to the reviewers comments. If you will need more time than this to complete your revisions, please reply to this message or contact the journal office at plosone@plos.org . Please include the following items when submitting your revised manuscript:

We look forward to receiving your revised manuscript.

Kind regards,

Md. Feroz Kabir, BPT, MPT, MPH, BPED, MPED

Academic Editor

PLOS ONE

2. We note that your Data Availability Statement is currently as follows: [All relevant data are within the manuscript and its Supporting Information files.] Please confirm at this time whether or not your submission contains all raw data required to replicate the results of your study. Authors must share the “minimal data set” for their submission. PLOS defines the minimal data set to consist of the data required to replicate all study findings reported in the article, as well as related metadata and methods (https://journals.plos.org/plosone/s/data-availability#loc-minimal-data-set-definition).

3. We note that Figure 2 in your submission contain [map/satellite] images which may be copyrighted. All PLOS content is published under the Creative Commons Attribution License (CC BY 4.0), which means that the manuscript, images, and Supporting Information files will be freely available online, and any third party is permitted to access, download, copy, distribute, and use these materials in any way, even commercially, with proper attribution. For these reasons, we cannot publish previously copyrighted maps or satellite images created using proprietary data, such as Google software (Google Maps, Street View, and Earth). For more information, see our copyright guidelines: http://journals.plos.org/plosone/s/licenses-and-copyright.

1. You may seek permission from the original copyright holder of Figure 2 to publish the content specifically under the CC BY 4.0 license. 

Additional Editor Comments:

Please send the revised papers according to the comments of the reviewers within one month.

Reviewers' comments:

Reviewer's Responses to Questions

**Comments to the Author**

1. Is the manuscript technically sound, and do the data support the conclusions?

Reviewer #1: Yes

Reviewer #2: Partly

Reviewer #3: Yes

2. Has the statistical analysis been performed appropriately and rigorously? 

Reviewer #1: Yes

Reviewer #2: No

Reviewer #3: Yes

3. Have the authors made all data underlying the findings in their manuscript fully available?

Reviewer #1: Yes

Reviewer #2: No

Reviewer #3: Yes

4. Is the manuscript presented in an intelligible fashion and written in standard English?

Reviewer #1: Yes

Reviewer #2: Yes

Reviewer #3: Yes

5. Review Comments to the Author

Reviewer #1: Thank you for this opportunity to revise the manuscript titled "Death Toll among the Bangladeshi Refugees of the 1971 War" that was submitted to PLOS ONE.

The urgency and timeliness of the research, realized by the authors of the reviewed article, are obvious. The provided facts have evident theoretical importance. The paper is characterized by a consistent summary of the content and accurate understanding of the investigated object. The conclusions, which are made by the authors, are naturally inferred from the results of the research.

This paper holds actual value to the readers on PLOS ONE.

I believe that the Research Article "Death Toll among the Bangladeshi Refugees of the 1971 War" corresponds to all the requirements, and it is sure to be recommended for publication.

Reviewer #2: The manuscript entitled "Death toll among the Bangladeshi refugees of the 1971 war" investigates the mortality rates among the aforementioned refugees during the 1971 liberation war. The authors rely on a binomial distribution model to analyze the mortality data. The approach may be methodological sound, however, there are issues with the study. In particular, the study focused on refugees camps, which would not account for deaths occurring outside these camps, and therefore underestimates the overall death toll. Additionally, study was not placed in other comparative analysis of crises to contextualize their findings and enhance the value of their analysis.

Reviewer #3: The authors are requested to address the below feedback which includes corrections and clarifications:-

P2, Abstract, sentence 2 – sentence should not start with a figure. Rewrite as follows: To avoid persecution by the Pakistani army, 9.9 million Bangladeshi refugees escaped to India.

P4, paragraph 3, sentence 3 – this sentence is written incorrectly. Rewrite sentence as follows: However, analyses of the death toll in the 1971 liberation war is usually confined to killings or violent deaths during the period.

P4, para 3, sentence 5: this sentence is also written incorrectly. Rewrite as follows: Even more neglected are the mortality records for the 9.9 million refugees who moved to India in 1971.

P5, para 1, last sentence – correct as follows: change Late to late (lower case L). Put year of interview after November-December.

P5, para 2, last sentence – delete govt. Spell out in full: government.

P5, para 3, first sentence – amend as follows: Seaman (1972) performed a .....

P5, para 3, third sentence – inconsistent style. Change two hundred thousand to 200,000.

P5, para 3, last sentence – extrapolated is not the correct word, use different word. Change I to lower case as follows: For example, in a news....

P6 – combine second and third paragraphs into one.

P7 and in all later pages – spell out Govt. in full, wherever it is used.

P18, para 1 - the total variance of 6,000,036,589.83 is an unrealistic figure as the total population of India in 1971 was about 547.9 million, and the total population of Bangladesh in 1971 was about 70 million.

Please provide explanation as to why the formula provides an unrealistic total variance of 6,000 million, which is many many times greater than the population of India and Bangladesh. It is also many times greater than the 9.9 million refugees who moved to India in 1971.

P18, last sentence - variance of the excess death toll, before adjustment = 6,000,122,924.29. Please provide explanation as to why the formula provides an unrealistic total variance of more than 6,000 million, which is much greater than the populations of India and Bangladesh. It is much higher than the 9.9 million refugees who moved to India in 1971.

P23 – it is mentioned that the refugees were largely Hindus. Is it possible to provide a breakdown of the religion of the 9.9 million refugees who moved to India in 1971?

P23, last 2 lines – the figures are written in this style: “562 thousand, .... between 323 and 802 thousand deaths”. But in other parts of the article and the conclusion, they are written in full figure style. Be consistent with style use.

P24, last sentence – use the word lakhs in the figures in the last sentence is not consistent with the style used in the article (5.6 lakhs, 0.5, 1 or 3 lakhs). Use the same style as the rest of the article – that is, write figures in thousands and millions.

The use of the word conservative for estimate of excess refugee death, is incorrect. Change to appropriate word as the estimate is higher and therefore not conservative.

The use of word easily refutes the low estimates is incorrect, as that is a subjective view of the author. Delete the word ‘easily’.

6. PLOS authors have the option to publish the peer review history of their article (what does this mean? ). If published, this will include your full peer review and any attached files.

**Do you want your identity to be public for this peer review?** For information about this choice, including consent withdrawal, please see our Privacy Policy .

Reviewer #1: **Yes: ** Prof. Dr. Vsevolod Konstantinov

Reviewer #2: No

Reviewer #3: No

---

## [Author Response · Author response to Decision Letter 0]

17 Feb 2025

Responses to reviewers' comments to the Author:

Reviewer #1:

Thank you for this opportunity to revise the manuscript titled "Death Toll among the Bangladeshi Refugees of the 1971 War" that was submitted to PLOS ONE.

The urgency and timeliness of the research, realized by the authors of the reviewed article, are obvious. The provided facts have evident theoretical importance. The paper is characterized by a consistent summary of the content and accurate understanding of the investigated object. The conclusions, which are made by the authors, are naturally inferred from the results of the research.

This paper holds actual value to the readers on PLOS ONE.

I believe that the Research Article "Death Toll among the Bangladeshi Refugees of the 1971 War" corresponds to all the requirements, and it is sure to be recommended for publication.

We thank the reviewer for their encouraging assessment. The reviewer did not appear to suggest specific corrections to implement.

Reviewer #2:

The manuscript entitled "Death toll among the Bangladeshi refugees of the 1971 war" investigates the mortality rates among the aforementioned refugees during the 1971 liberation war. The authors rely on a binomial distribution model to analyze the mortality data. The approach may be methodological sound, however, there are issues with the study. In particular, the study focused on refugees camps, which would not account for deaths occurring outside these camps, and therefore underestimates the overall death toll. Additionally, study was not placed in other comparative analysis of crises to contextualize their findings and enhance the value of their analysis.

We thank the reviewer for their summary of the manuscript and suggesting that the approach may be methodological sound.

Regarding the reviewer’s point about deaths occurring outside these camps, the manuscript already acknowledges this as one of the limitations under the section ‘Why our estimate of death toll is likely to be an underestimate’. Furthermore, a detailed review of literature about deaths occurring outside these camps is provided in supplementary information S9 and S10 Text. We acknowledge this limitation and explain that due to a paucity of data this limitation cannot be addressed statistically at present.

Regarding the reviewer’s point about other comparative analysis of crises, we believe that we have already done so in our manuscript in the section ‘Comparison to estimated death rates in other refugee crises around the world’, where we discuss published reviews on estimated death rates in other refugee crises around the world, for example, by citing Heudtlass, Speybroeck, & Guha-Sapir, 2016; Toole & Waldman, 1988; etc. We place our estimated CDR and death rate ratio in context of those estimated in other refugee crises by those studies, showing that our estimates are comparable to the range of estimates observed in other crises.

The reviewer did not appear to suggest specific corrections to implement.

Reviewer #3:

The authors are requested to address the below feedback which includes corrections and clarifications:-

We thank the reviewer for their positive assessment of the manuscript and the suggestions for corrections.

P2, Abstract, sentence 2 – sentence should not start with a figure. Rewrite as follows: To avoid persecution by the Pakistani army, 9.9 million Bangladeshi refugees escaped to India.

This has been implemented.

P4, paragraph 3, sentence 3 – this sentence is written incorrectly. Rewrite sentence as follows: However, analyses of the death toll in the 1971 liberation war is usually confined to killings or violent deaths during the period.

This has been implemented.

P4, para 3, sentence 5: this sentence is also written incorrectly. Rewrite as follows: Even more neglected are the mortality records for the 9.9 million refugees who moved to India in 1971.

This has been implemented.

P5, para 1, last sentence – correct as follows: change Late to late (lower case L). Put year of interview after November-December.

This has been implemented.

P5, para 2, last sentence – delete govt. Spell out in full: government.

This has been spelled out in full throughout the manuscript.

P5, para 3, first sentence – amend as follows: Seaman (1972) performed a .....

This has been amended to “Seaman performed a survey on a sample population of 4,770 refugees in the Salt Lake refugee camp [25] and…” since the citation formatting guidelines now require us to use numerical citations.

P5, para 3, third sentence – inconsistent style. Change two hundred thousand to 200,000.

This has been implemented. Similar changes to numbers have been implemented throughout the manuscript, such as ‘seventy million’ to ‘70 million’.

P5, para 3, last sentence – extrapolated is not the correct word, use different word. Change I to lower case as follows: For example, in a news....

Both changes have been implemented. The amended text now says, “For example, in a news report on 25th December 1971, Seaman himself mentioned that in the Salt Lake camp, 6,000 children aged below eight are estimated to have died out of a total population of 31,000; this death rate applied to the total refugee children population would mean 300,000 deaths among the children aged under eight alone [29].”

P6 – combine second and third paragraphs into one.

This has been implemented.

P7 and in all later pages – spell out Govt. in full, wherever it is used.

This has been spelled out in full throughout the manuscript.

P18, para 1 - the total variance of 6,000,036,589.83 is an unrealistic figure as the total population of India in 1971 was about 547.9 million, and the total population of Bangladesh in 1971 was about 70 million.

Please provide explanation as to why the formula provides an unrealistic total variance of 6,000 million, which is many many times greater than the population of India and Bangladesh. It is also many times greater than the 9.9 million refugees who moved to India in 1971.

To clarify, while the approximation formula for the classical binomial distribution would appear to indicate that the numerical values of the mean and variance are similar, it is important to note that they have different units, and therefore are not directly comparable. While the mean is in the unit of ‘number of deaths’, the variance is in the unit of (number of deaths)2. Only its square root, the standard deviation, comes back to the same unit, number of deaths.

To illustrate this difference, if we were to use the unit of ‘millions of deaths’ instead, the estimated total death toll would be 649,249.40 / 1,000,000 = 0.649 millions of deaths. But with this unit, the estimated variance would be 6,000,036,589.83 / 1,000,0002 = 0.006 (millions of deaths)2, which would now appear numerically far smaller than the estimated death toll. Whereas the estimated standard deviation, its square root, will be 0.077 millions of deaths, comparable to the estimated death toll of 0.649 millions of deaths, although one unit of magnitude smaller. This should clarify why the large numerical magnitude of the variance should not be an issue of concern.

P18, last sentence - variance of the excess death toll, before adjustment = 6,000,122,924.29. Please provide explanation as to why the formula provides an unrealistic total variance of more than 6,000 million, which is much greater than the populations of India and Bangladesh. It is much higher than the 9.9 million refugees who moved to India in 1971.

We hope that the clarification provided above also explains this question around the magnitude of variance.

P23 – it is mentioned that the refugees were largely Hindus. Is it possible to provide a breakdown of the religion of the 9.9 million refugees who moved to India in 1971?

As referred to in that sentence, some further information has been provided in supplementary S4 text about the religious composition of the refugee population, to the extent available from various reports.

“(Chen & Rohde, 1973, p. 199) explained the official stance by suggesting “some have attributed this distortion to the fact that Pakistan did not recognize the Hindu refugees (who numbered about 8 million) as legitimate citizens of the Muslim state of Pakistan.” … These suggestions are likely based on the observations that Hindus constituted the majority of the refugee population, estimated to be between 70-90% (Foreign Relations of the United States, U. S. State Department, 1971) (Saha, 2003, p. 217) (Gerlach, 2012, p. 137)”

However no direct survey seems to have been conducted on the religious composition of the refugee population or among the camps, and the sources do not provide further information on how these estimates were obtained. As the calculations in our manuscript did not depend on the religion of the refugees, we did not feel the need of going into much further detail about religion in the main text; we thus put the relevant literature review in supplementary text.

P23, last 2 lines – the figures are written in this style: “562 thousand, .... between 323 and 802 thousand deaths”. But in other parts of the article and the conclusion, they are written in full figure style. Be consistent with style use.

This has been implemented.

P24, last sentence – use the word lakhs in the figures in the last sentence is not consistent with the style used in the article (5.6 lakhs, 0.5, 1 or 3 lakhs). Use the same style as the rest of the article – that is, write figures in thousands and millions.

We wanted to provide the estimates in the unit of lakhs in the concluding sentence of the manuscript, because lakh is the main numerical multiple of thousand used as a unit in South Asia including Bangladesh, where million is not a common unit. Therefore, the majority of the discourse around refugee numbers and deaths were and continue to be in lakhs, e.g. in the books and news articles used in this study, such as Saha, 2003. That book, for example, quotes the Indian Prime Minister Indira Gandhi as “15 lakh people from Bangladesh had come to India to protect themselves from the atrocities of Pakistani forces.” Therefore, we wanted to make our final estimate easily understandable and comparable for a general reader from South Asia, by quoting it in lakhs in addition to millions.

The last sentence has now been modified to

“Therefore, our conservatively calculated estimate of excess refugee death – 5.6 lakhs (0.56 million; 1 million is equal to 10 lakhs) among roughly ten million refugees – refutes the low estimates such as 0.5, 1 or 3 lakhs (0.05, 0.1, or 0.3 million) for the total wartime death toll among the seventy million population.”

The use of the word conservative for estimate of excess refugee death, is incorrect. Change to appropriate word as the estimate is higher and therefore not conservative.

In the section ‘Why our estimate of death toll is likely to be an underestimate’ we explain why our estimate is conservative given the data organised in this study, as during construction of the estimate, we made conservative decisions in each of those steps. We now clarify that the word ‘conservative’ is specifically referring to our calculations, in two places of the manuscript. In this last sentence we rephrase it as “Therefore, our conservatively calculated estimate of excess refugee death…”

The use of word easily refutes the low estimates is incorrect, as that is a subjective view of the author. Delete the word ‘easily’.

We have deleted the word ‘easily’.

---

## [Editor Report · Decision Letter 1]

25 Feb 2025

Death Toll among the Bangladeshi Refugees of the 1971 War

PONE-D-24-42984R1

Dear Kaustubh Adhikari,

We’re pleased to inform you that your manuscript has been judged scientifically suitable for publication and will be formally accepted for publication once it meets all outstanding technical requirements.

Kind regards,

Md. Feroz Kabir, BPT, MPT, MPH, BPED, MPED

Academic Editor

PLOS ONE
---

## [Editor Report · Acceptance letter]

PONE-D-24-42984R1

PLOS ONE

Dear Dr. Adhikari,

I'm pleased to inform you that your manuscript has been deemed suitable for publication in PLOS ONE. Congratulations! Your manuscript is now being handed over to our production team.

Kind regards,

on behalf of

Dr. Md. Feroz Kabir

Academic Editor

PLOS ONE